# Methane emissions from oil and gas platforms in the North Sea

Stuart N. Riddick[1], Denise L. Mauzerall[1,2] Michael Celia[1], Neil R. P. Harris[3], Grant Allen[4], Joseph Pitt[4], John Staunton-Sykes[5], Grant L. Forster[6], Mary Kang[7], David Lowry[8], Euan G. Nisbet[8] and Alistair J. Manning[9]

[1] Department of Civil and Environmental Engineering, Princeton University, Princeton, 08544, USA
[2] Woodrow Wilson School of Public and International Affairs, Princeton University, Princeton, 08544, USA
[3] Centre for Environmental and Agricultural Informatics, Cranfield University, Cranfield, MK43 0AL, UK
[4] Centre for Atmospheric Science, University of Manchester, Manchester, M13 9PL, UK
[5] Centre for Atmospheric Science, University of Cambridge, Cambridge, CB2 1EW, UK
[6] National Centre for Atmospheric Science, University of East Anglia, Norwich, NR4 7TJ, UK
[7] Department of Civil Engineering and Applied Mechanics, McGill University, Montreal, H3A 0C3, Canada
[8] Department of Earth Sciences, Royal Holloway, University of London, Egham, TW20 0EX, UK
[9] Met Office, Exeter EX1 3PB, UK

*Correspondence to*: Stuart N. Riddick (sriddick@princeton.edu) and Denise L. Mauzerall (mauzeral@princeton.edu)

**Abstract.**

Since 1850 the concentration of atmospheric methane ($CH_4$), a potent greenhouse gas, has more than doubled. Recent studies suggest that emission inventories may be missing sources and underestimating emissions. To investigate whether offshore oil and gas platforms leak $CH_4$ during normal operation, we measured $CH_4$ mole fractions around eight oil and gas production platforms in the North Sea which were neither flaring gas nor off-loading oil. We use the measurements from summer 2017, along with meteorological data, in a Gaussian plume model to estimate $CH_4$ emissions from each platform. We find $CH_4$ mole fractions of between 11 and 370 ppb above background concentrations downwind of the platforms measured, corresponding to a median $CH_4$ emission of 6.8 g $CH_4$ s$^{-1}$ for each platform, with a range of 2.9 to 22.3 g $CH_4$ s$^{-1}$. When matched to production records, during our measurements individual platforms lost between 0.04% and 1.4% of gas produced with a median loss of 0.23%. When the measured platforms are considered collectively, (i.e. the sum of platforms' emission fluxes weighted by the sum of the platforms' production), we estimate the $CH_4$ loss to be 0.19% of gas production. These estimates are substantially higher than the emissions most recently reported to the National Atmospheric Emission Inventory (NAEI) for total $CH_4$ loss from United Kingdom platforms in the North Sea. The NAEI reports $CH_4$ losses from the offshore oil and gas platforms we measured to be 0.13% of gas production, with most of their emissions coming from gas flaring and offshore oil loading, neither of which were taking place at the time of our measurements. All oil and gas platforms we observed were found to leak $CH_4$ during normal operation and much of this leakage has not been included in UK emission inventories. Further research is required to accurately determine total $CH_4$ leakage from all offshore oil and gas operations and to properly include the leakage in national and international emission inventories.

# 1 Introduction

Methane (CH₄) is a greenhouse as well as a precursor of tropospheric ozone, which is widely regulated as a component of photochemical smog. Since 1850 atmospheric $CH_4$ mixing ratios have increased from 715 ppb to 1865 ppb in February 2019 with an annual increase of 10ppb/year in 2018 (NOAA, 2019). This increase is largely driven by anthropogenic activities though uncertainties exist in the magnitude of individual source sectors and sinks (Turner et al., 2019). Observatories have been set up around the world to track trends in $CH_4$ concentrations (de Coninck et al., 2018).

Between 2012 and 2015 unusually high $CH_4$ enhancements of up to 400 ppb above background were observed at the University of East Anglia's Weybourne Atmospheric Observatory (WAO; 52.95 °N, 1.14 °E) during periods of northerly onshore winds and high surface pressures (Connors, 2015; Staunton-Sykes, 2016; Connors et al., 2018). These elevated enhancements were unexpected as the air came from the open ocean. However, a potential source of these $CH_4$ enhancements is leakage from offshore oil and gas production platforms 80 km away from WAO in the North Sea (OSPAR, 2018; Fig. 1).

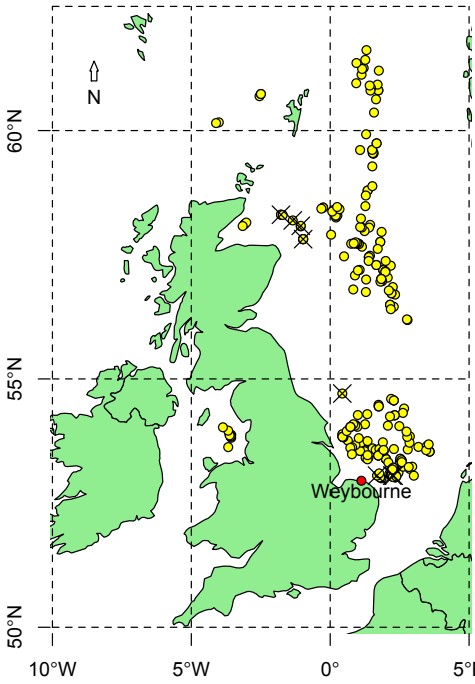

**Figure 1 Map of the North Sea showing the locations of all UK offshore oil/gas platforms (the filled yellow circles) and the eight platforms measured by this study (black crosses) (Source: OSPAR, 2012). The map also shows the location of the University of East Anglia's Weybourne Atmospheric Observatory (WAO; 52.95 °N, 1.14 °E) in Weybourne, Norfolk, UK.**

In 2015 the UK extracted about 32 Tg of natural gas from the North Sea (UK Oil and Gas Authority, 2018). During this time, a loss of 40 Gg $CH_4$ (0.13 % of natural gas production) was reported by the UK Government Department for Business, Energy & Industrial Strategy (BEIS), mainly through venting (24 Gg $CH_4$ yr⁻¹) and flaring (12 Gg $CH_4$ yr⁻¹) activities (BEIS, 2018). However, recent studies indicate public inventories in the United States underestimate $CH_4$ emissions including from the oil

and gas supply chain (Alvarez et al., 2018; Zavala-Araiza et al., 2015, Schwietzke et al., 2017). This leads to the question: Could $CH_4$ emissions from offshore oil and gas platforms be higher than previously estimated?

Land-based measurements in West Virginia and Colorado, USA, estimate that onshore oil and gas extraction activities lose between 0.1 and 10 % of $CH_4$ produced (Petron et al., 2012, Omara et al., 2016; Schwietzke et al., 2017; Alvarez et al., 2018; Englander et al., 2018; Riddick et al., 2019). At present there are no direct, near-source measurements of $CH_4$ emissions from offshore oil and gas production. However, a mass-balance approach identifies $CH_4$ emissions from offshore oil and gas operations off the coast of South East Asia as having a large regional median (range) emission of 99 (4 – 427) g $CH_4$ $s^{-1}$ $platform^{-1}$ for the Malay Peninsula and 15 (2 – 46) g $CH_4$ $s^{-1}$ $platform^{-1}$ for Borneo (Nara et al., 2015). Numerous productive offshore oil and gas fields exist across the globe, including Saudi Arabia, Brazil, Mexico, Norway and the United States, making careful measurement and analysis of leakage from these platforms important for global emission inventories.

Several activities on offshore production platforms in the North Sea are explicitly identified by the BEIS in the National Atmospheric Emission Inventory (NAEI) as sources of $CH_4$ emission including: combustion activities such as gas flaring, offshore oil loading and venting directly to the atmosphere (BEIS, 2018). Leakage during normal operations are not explicitly included. Oil and gas operators report an annual $CH_4$ emission estimate for each offshore production platform to the NAEI; these emission estimates are primarily calculated using emission factors (Butterfield, 2017). Technical guidance on the emission factor based calculations is available through the UK Government's Department of Energy and Climate Change Environmental and Emissions Monitoring System (DECC EEMS, 2008). The main shortcoming of using emission factors and activity levels to estimate total emissions is that total emissions can be underestimated if not all emission sources are identified. For example, leaks not obvious to platform personnel would not be included and the total emission would be an underestimate of $CH_4$ lost. Overall, as emission factor calculations rely on explicit knowledge of all sources of leakage, current approaches used by industry could underestimate total $CH_4$ emissions from offshore installations.

In this study we investigate $CH_4$ emissions from offshore oil and gas installations in the North Sea and determine how they differ from those currently reported by the BEIS. To investigate the $CH_4$ loss from offshore oil and gas installations in UK waters, we measure $CH_4$ mixing ratios downwind of offshore platforms and use these data in a Gaussian plume model to estimate $CH_4$ emission rates. The $CH_4$ loss is then presented as a percentage of the $CH_4$ produced by each platform.

## 2 Methods - Boat based observations

Oil and gas platforms in UK waters are located between 30 and 500 km from the UK mainland, with the majority of platforms located to the east of the UK in the North Sea (Fig. 1; OSPAR, 2018). To investigate possible emissions from these platforms, sea-level $CH_4$ mole fractions were measured around eight oil and gas platforms between the 6th June and 25th August 2017. Measurements were made during normal operation (i.e. pilot light on the flare stack was lit, but no flaring or offshore oil loading was observed). Where possible a full circle was made around the installation to observe the upwind and downwind methane mole fractions. To determine if flaring or offshore oil loading was occurring, a visual inspection was made of the

installation. We assumed that venting was not taking place because no venting was reported in any of the most recent NAEI. Previously published emissions from the measured platforms in the NAEI are reported to be almost entirely due to flaring (83%) and offshore oil loading (17%), with reported emissions generated using emissions factors (Brown et al., 2017; BEIS, 2018).

The oil and gas platforms measured here were selected at random, constrained only by the need for accessibility. Fishing boats were chosen as the measurement platforms because of budgeting and availability constraints. Maritime and Coastguard Agency regulations for the available vessels (MCA category 2) meant that the platforms measured had to be less than 60 miles from a safe haven. Four of the eight platforms only produced natural gas (#1 to #4) that was transported to the mainland via pipeline, while the remaining four produced oil and gas. Two of the oil and gas platforms (#5 and #6) include floating
production storage and offloading vessels, which receive hydrocarbons, process them and store them until they can be offloaded by tanker or pipeline, and the other two platforms (#7 and #8) transport oil and gas directly to the mainland by pipeline. Methane mole fractions, latitude, longitude, and meteorological data were collected as the boat travelled upwind and downwind of the platforms.

## 2.1 Methane mole fraction measurements – Los Gatos UGGA

The Los Gatos Research Ultra-portable Greenhouse Gas Analyzer (UGGA; www.lgrinc.com) was used to measure $CH_4$ concentrations near the off-shore oil and gas platforms. The UGGA is a laser absorption spectrometer that measures $CH_4$ mole fractions in air (Paul et al., 2001). The UGGA reports $CH_4$ mole fractions every second, with a stated precision of < 2 ppb ($1\sigma$ @ 1 Hz) over an operating range of 0.1 to 100 ppm. Calibration of the UGGA was conducted before and after deployment using low (1.93 ppm $CH_4$), target (2.03 ppm $CH_4$) and high (2.74 ppm $CH_4$) mole fraction gases calibrated on the World
Meteorological Organization (WMO) scale. Measurements were taken between the edge of the exclusion zone (500 m from the platform; HSE, 2018) and 2 km horizontal distance from the platforms. The inlet line was attached to a mast 2.5 m above sea level, to avoid contamination from the boat's exhaust, and protected from water incursion using an aluminium funnel. The air was filtered using a 2 μm filter. Background $CH_4$ mole fractions were measured while the boat was upwind of the production platform.

## 25  2.2 Meteorological data

Meteorological data were collected using a wireless weather station (Maplin, UK) attached to a mast 2 m above sea level. Data were sampled and recorded at one-minute intervals and included: wind speed ($u$, m s$^{-1}$), wind direction ($WD$, ° to North), air temperature at 2 m ($T_a$, K), relative humidity ($RH$, %), rain rate ($R$, mm hr$^{-1}$), irradiance ($I$, W m$^{-2}$) and air pressure ($P$, Pa). The wind speed used in the emission modelling was corrected for emission height using a wind profile power law (Touma,
1977; Hsu et al., 1994).

## 2.3 Gaussian Plume Model

The Gaussian plume model used in this study calculates the mole fraction of a gas as a function of distance downwind from a point source (Seinfeld and Pandis, 2006). As a gas is emitted, it is entrained in the prevailing ambient air flow and disperses in the $y$ and $z$ directions (relative to a mean horizontal flow in the x direction) with time, forming a cone. The mole fraction of the gas as a function of distance downwind depends on the emission flux at the source, the advective wind speed ($u$, m s$^{-1}$) and the rate of dispersion. The mole fraction of the gas ($X$, µg m$^{-3}$), at any point $x$ meters downwind of the source, $y$ meters laterally from the centre line of the plume, and $z$ metres above ground level can be calculated (Eq. 1) using the source strength ($Q$, g s$^{-1}$), the height of the source ($h_s$, m), the height of the boundary layer ($h$, m) and the stability of the air (CERC, 2017; Hunt, 1982; Hunt et al., 1988). The standard deviation of the lateral ($\sigma_y$, m) and vertical ($\sigma_z$, m) mixing ratio distributions are calculated from the Pasquill Gifford stability class (PGSC) of the air (Pasquill, 1962; Busse and Zimmerman, 1973; US EPA, 1995). Even though this modelling method is relatively simple, offshore emissions estimates using the same parameterization of $\sigma_y$ and $\sigma_z$ were made by Blackall et al. (2008) and were in good agreement ($R^2 = 0.85$) with emissions calculated from a concurrent tracer release experiment. Alternative offshore parameterisations for $\sigma_y$ and $\sigma_z$ exist and are used in the EPA recommended Offshore and Coastal Dispersion Model (Hanna et al, 1985). However, these algorithms require further data on the micrometeorology which are not available and were therefore not used as they introduce additional unquantifiable uncertainty.

$$X(x,y,z) = \frac{Q}{2\pi u \sigma_y \sigma_z} e^{\frac{-y^2}{2\sigma_y^2}} \left( e^{\frac{-(z-h_S)^2}{2\sigma_z^2}} + e^{\frac{-(z+h_S)^2}{2\sigma_z^2}} + e^{\frac{-(z-2h+h_S)^2}{2\sigma_z^2}} + e^{\frac{-(z+2h-h_S)^2}{2\sigma_z^2}} + e^{\frac{-(z-2h-h_S)^2}{2\sigma_z^2}} \right) \qquad (1)$$

The following assumptions are made regarding the Gaussian model: 1) The source is emitting CH$_4$ at a constant rate; 2) The mass of CH$_4$ is conserved when reflected at the surface of the ocean or the top of the boundary layer; 3) Wind speed and vertical eddy diffusivity are constant with time; 4) There is uniform vertical mixing; and 5) Terrain (ocean surface) is relatively flat between source and detector. The PGSC were determined for an offshore flow of air following the parametrizations described in Erbrink and Scholten (1995), Hanna et al. (1985) and Hsu (1992).

## 2.4 Gaussian Plume model parameterization

A Gaussian plume approach was used to infer the CH$_4$ emissions flux from the oil and gas platforms using the CH$_4$ mole fraction data collected downwind. We used measurements for the mole fraction, and rearranging Equation 1 solved for the source term Q. Data used as input to the Gaussian plume model are: wind speed, wind direction, temperature, minute-averaged CH$_4$ mole fraction at 2 m, background CH$_4$ mole fraction and the PGSC. For the minute-averaged CH$_4$ mole fraction data, we assume the 1-minute averaged data near the centre of the observed instantaneous plume is representative of the centre of the time-averaged Gaussian plume. The PGSC are estimated from wind speed and irradiance data (Turner, 1970; Seinfeld and

Pandis, 2006), as measured by the meteorological station on the boat. The height of the boundary layer is calculated from the Global Forecast System's global forecast model archives (GFS, 2019).

An unknown variable used in the Gaussian plume model in this study is the height at which emissions are released. The emissions could have come from the working deck of the platform, the top of the flare or somewhere in between. For the purposes of the emission estimates calculated and presented here, we assume $CH_4$ is emitted from the working deck only which results in the smallest emissions possible for a given measurement. As a sensitivity study, emissions were also calculated assuming the source was at the top of the flare stack only (see Supplementary Material Section 1). The height of the working deck and the height of the flare stack at each of the platforms were determined using platform characteristics data from each oil and gas platform available on the internet.

## 2.5 Uncertainties

Of the Gaussian plume model assumptions presented in Section 2.3, two may not be valid - uniform vertical mixing and a constant wind speed. The uncertainty in uniform vertical mixing is discussed in Section 3.4. To investigate how uncertainties in the measurements and modelling affect the calculated emission, we ran Gaussian plume model scenarios using data that reflect the input values' uncertainty bounds. The scenarios run using the Gaussian plume approach were: varying wind speed (based on measurement); UGGA precision ($\pm$ 2 ppb); thermometer precision ($\pm$ 0.1 °C); the PGSC (+ 1 PGSC); and distance from detector to emission source ($\pm$ 50 m). The uncertainties of the UGGA and thermometer were taken from literature. The uncertainty in the PGSC used reflects the possibility that the temperature of the natural gas leaving the subsurface could be hotter than air and therefore less stable. The uncertainty in distance from the emission source to the detector results from not knowing where gas is leaking; here we assume the leak could be from anywhere on a production platform that is 100 m long.

## 2.6 Data sources

The UK Department for Business, Energy & Industrial Strategy (BEIS) keeps the Environmental and Emissions Monitoring System (EEMS) which is the environmental database of the UK oil and gas industry. Methane emission data are uploaded to this by industry partners. These data form the basis for emissions reported under category 1B2 within the National Atmospheric Emissions Inventory (NAEI; BEIS, 2018). For details of how this data is incorporated into the NAEI, see Brown et al. (2017). The most recent point-source emission database from the NAEI available at the time of writing was for the year 2015. Individual platform production data for both 2015 and 2017 were taken from the Petroleum Production Reporting system published by the UK Oil and Gas Authority (OGA, 2018).

# 3 Results

## 3.1 Methane mole fractions around North Sea oil and gas platforms

Our sea-level surveys indicate $CH_4$ mole fraction enhancements can be measured near all of the production platforms observed, when upwind $CH_4$ mole fractions ($[CH_4]_{bgd}$, ppb) are compared with downwind mole fractions ($[CH_4]$, ppb; Table 1). The

largest enhancement of 370 ppb was observed downwind of Platform #6 on the 24[th] August 2017, while the lowest enhancement of 11 ppb was observed downwind of platform #8 on the 25[th] August 2017. The median $CH_4$ enhancement downwind of the eight platforms was 43 ppb (mean: 112 ppb; range: 11 – 370 ppb). While measurements were being conducted, a maximum variability in wind speed of $\pm 0.6$ m s$^{-1}$ was measured at platform #3 on the 6[th] July 2017; during no measurements did an observable change in wind direction occur. Complete circles of all installations were not possible due

to access restrictions, i.e. the measurement vessel could not get between some platforms and maintain the 500 m clearance required of each platform, and there were occasions when the measurement boat was actively blocked by the platform's standby vessel (Supplementary Material Section 2).

**Table 1. Meteorological, position and mole fraction data taken during the boat-based measurement campaign around oil and gas production platforms in the North Sea between 6[th] June and 25[th] August 2017. Emissions were calculated assuming the source of**
**the emissions was at the working deck level. The calculation of the "Median", "Mean" and "Total" only use data from platforms #4 to # 8. Platforms #1 and #2 did not have production data available for the time of measurement. During the measurement of Platform #3 the height of the PBL was calculated as zero (GFS, 2019) making the Gaussian plume modelled emission estimate ambiguous.**

| ID # | Measurement Date | Type | Height (m) | Peak enhancement at the centre of the plume (ppb) | Emission (g s$^{-1}$) | Platform $CH_4$ production (g s$^{-1}$) | Loss (%) |
|---|---|---|---|---|---|---|---|
| 1 | Jun 6 | Gas | 40 | 50 | 4.9 | N/A | |
| 2 | Jun 6 | Gas | 40 | 51 | 4.9 | N/A | |
| 3 | Jul 5 | Gas | 40 | 34 | 1.1 | 672 | 0.17 |
| 4 | Aug 16 | Gas | 50 | 30 | 5.7 | 15,230 | 0.04 |
| 5 | Aug 24 | Oil/Gas | 50 | 370 | 22.3 | 1,585 | 1.41 |
| 6 | Aug 24 | Oil/Gas | 50 | 312 | 18.1 | 1,845 | 0.98 |
| 7 | Aug 25 | Oil/Gas | 50 | 35 | 6.8 | 2,952 | 0.23 |
| 8 | Aug 25 | Oil/Gas | 50 | 11 | 2.9 | 8,047 | 0.04 |
| | | | | Median   (#4 - #8) | 6.8 | | 0.23 |
| | | | | Mean     (#4 - #8) | 11.2 | | 0.54 |
| | | | | Total     (#4 - #8) | 55.8 | 29,662 | |
| | | | | Loss of $CH_4$ produced (%)      (#4 - #8) | 0.19 | | |

## 3.2 Source of leaks

Although the production platforms measured in this campaign were not actively flaring gas, (i.e. burning gas to reduce pressure during oil extraction), the pilot light on the top of the flare stack was actively burning gas. As an example, Fig.2 shows the

minute-averaged $CH_4$ enhancements upwind and downwind of a production platform on the 24th August. This example was chosen as it was the only platform that had an offset flare stack (i.e. not centred in the platform). Fig.2 indicates the largest enhancement was downwind of the flare stack. The width of the plume (< 200 m) suggests a compact $CH_4$ source. This could be associated with incomplete combustion of natural gas feeding the pilot light at the top of the platform, or it could be associated with gas being emitted at the working deck level.

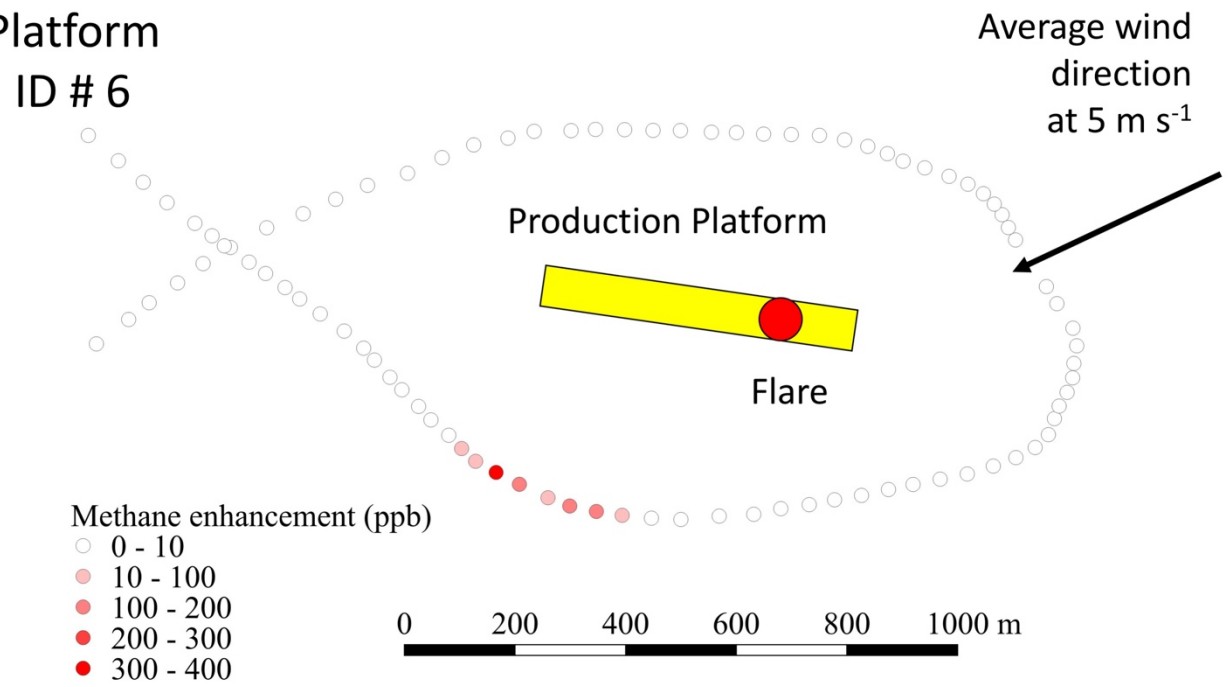

**Figure 2. Minute-averaged $CH_4$ enhancements made upwind and downwind of production platform # 6, on the 24th of August 2017.**

### 3.3 Estimating methane emissions

Using Gaussian plume modelling and assuming all emissions came from the working deck, the highest emission of 22.3 g s$^{-1}$ $CH_4$ was observed from platform # 5 on the 24th August 2017 while the lowest emission, 2.9 g s$^{-1}$, was observed on the 25th August 2017 from platform # 8. During the measurement of platform #3, the calculated boundary layer height was 0 m (GFS, 2019) making the emission estimate ambiguous and, even though presented in Table 1, has not been used further in the analysis. Using emission data from the five platforms with available production data and with a non-zero calculated PBL (platforms #4 through #8), the median $CH_4$ emission was 6.8 g s$^{-1}$ (mean 11.2 g s$^{-1}$). As a sensitivity study, the median modelled emission is 2,658 g s$^{-1}$ (mean 1,892 g s$^{-1}$) when we assume all $CH_4$ is emitted from the highest point of the platform, i.e. the flare. When normalized against natural gas production data (OGA, 2018), the highest $CH_4$ loss rate corresponded to 1.4 % of production at platform #5 while the lowest loss rate corresponded to 0.04 % of production at platforms #4 and #8. We estimate

the median CH$_4$ loss from platforms #4 through #8 to be 0.23 % of production. When weighted by production, i.e. the collective emission from the measured platforms (56 g s$^{-1}$; Table 1) as a fraction of the collective production of the measured platforms (29,662 g s$^{-1}$; Table 1), the average loss from all measured platforms was 0.19 % of their total production.

For comparison, we have also calculated the reported loss rates for 2015 using the most recent NAEI emissions data (Brown et al., 2017; BEIS, 2018). We find the median reported loss rate from NAEI was 0.23 % for the six platforms we measured where production data was available, with a production-weighted average of 0.19 %. These values are close to those we calculated. However, this apparent consistency is misleading as the NAEI emissions are dominated by CH$_4$ emissions from flaring and offshore oil loading activities neither of which were occurring during our measurement periods; this is discussed further in section 4.

## 3.4 Uncertainties/shortcomings of Gaussian Plume modelling

A range of scenarios were run using the Gaussian plume model to estimate uncertainty in average CH$_4$ emissions resulting from UGGA instrument precision, thermometer precision, varying wind speed, assessment of the PGSC and uncertainty in distance between the emission source and the detector. Uncertainty in the UGGA and the thermometer have little effect on the average emission estimate (Supplementary Material Section 3). The largest variability in wind speed was recorded during measurement of platform # 3 on the 6$^{th}$ July 2017 at 4.4 ± 0.6 m s$^{-1}$ (Supplementary Material Section 1), using this variability in wind speed in the Gaussian plume model results in an uncertainty in average emission of ± 12 %. Uncertainty in estimating the distance between the emission source and the detector results in an uncertainty in average emissions of ± 8 %. The Gaussian plume model has the greatest response to the uncertainty in estimating the PGSC, resulting in an uncertainty of ± 41 %. We estimate the overall uncertainty in the average CH$_4$ emission, calculated as the root of the sum of the individual uncertainties squared, to be ± 45 %.

As mentioned in Section 2.5, the uniform vertical mixing assumption made in the Gaussian Plume model may not hold here as the data we collected provides no information on vertical mixing. However, the Gaussian plume model only assumes a constant vertical mixing rate between the source and the detector. In most cases this distance is relatively short and unlikely to significantly affect the calculation of emissions. In future experiments, the vertical mixing rate could be calculated by measuring the vertical gradient of wind speeds to make an accurate thermodynamic profile.

## 4 Discussion

From boat based observations we observed elevated CH$_4$ mole fractions, between 11 and 370 ppb above background, downwind of eight oil and gas production platforms in the North Sea when none of the platforms was engaged in either gas flaring or oil transfer and unloading. This suggests that all observed oil and gas platforms leak CH$_4$ during normal operations. Using the near-source CH$_4$ measurements in a simple Gaussian plume model (where the CH$_4$ emissions are calculated from the minute-averaged peak enhancement at the centre of the plume), we found the median of the calculated CH$_4$ emissions from

offshore oil and gas installations to be 589 kg $CH_4$ day$^{-1}$, with individual platform's $CH_4$ emissions ranging from 98 to 1,928 kg $CH_4$ day$^{-1}$. Matching production data to our measurements we estimate 1) a median loss of $CH_4$ from the six platforms, unweighted by production, of 0.23 % (mean 0.54 %) and 2) the cumulative loss of $CH_4$, weighted by total production, of 0.19%. These results indicate that, of the platforms measured, those producing more gas leaked proportionally less of what

they produced. Also, the two higher emitting platforms (#5 and #6) include floating production storage and offloading vessels; we find these to have much larger loss rates than the three fixed platforms (#4, #7 and #8). However, we also acknowledge our sample size is small and the five platforms may not be indicative of the overall performance of platforms in the North Sea. The 2015 emission factor based NAEI emissions are within the ranges calculated in this study, i.e. a median loss rate of 0.23% and a production-weighted loss of 0.19%, and also show larger losses come from lower producing platforms. However, the

NAEI provides the main source of emission for each installation and their reported emissions from the six platforms are almost entirely due to flaring (83%) and offshore oil loading (17%), neither of which was taking place during our measurements. Typically, these activities are not continuous on North Sea platforms; consequently, emission rates are likely to be much higher at certain times than others. As flaring and oil loading did not coincide with our measurement campaign, the measured emissions presented here represent leakage only and do not account for intermittent emissions due to venting, flaring or oil

loading activities. This suggests a potentially large missing source of $CH_4$ emissions in the national U.K. $CH_4$ emission inventory.

The emission estimates presented here are from a pilot study and further work is needed to establish total $CH_4$ leakage rates from offshore oil and gas platforms. We have established, however, that $CH_4$ enhancements can be detected downwind of all production platforms during normal operations when neither venting, flaring or oil loading activities are taking place. Our

measurements used in a Gaussian plume model indicate leakage from offshore installations are likely larger than previously estimated. However, these emission estimates come with large uncertainties as they are based on relatively few measured platforms, assume values for the height of emission, lateral and vertical mixing ratio distributions, and may not meet all the Gaussian plume model assumptions.

When the $CH_4$ emissions are calculated for two different emission heights, the importance of identifying the source location

and height above the sea becomes apparent. The median $CH_4$ emissions from the five platforms is 6.8 g s$^{-1}$ when the emissions are all assumed to come from the working deck, while the median emissions is 2,658 g s$^{-1}$ (47% of production) when all $CH_4$ is assumed to be emitted from the flare i.e. the highest point of the platform. This analysis indicates that the median emission presented here, based on the assumption that the emissions occur from the working deck, is a conservative estimate. However, without further measurements the height of the emission source cannot be definitively determined and this leaves the possibility

that leakage is higher during normal operations than our results indicate. The other input variables that cannot be determined without further measurement are the lateral and vertical mixing ratio distributions but we feel that following the study of Blackall et al. (2007) the estimates used in this study are sufficient to establish leakage from oil and gas platforms and to provide a rough estimate of their emissions. As with the emission height, mixing can be resolved with further measurement, including the use of aircraft to resolve the vertical and horizontal mixing of the plume.

It is clear that further studies are needed to provide additional data that will yield more definitive emission estimates. Using the near-source (< 1 km) observations of this paper (Fig. 2; Supplementary Material Section 2 platform #5 and #6) we can see that plumes from the leaks are compact (<200 m wide) and in some cases difficult to detect from sea-level measurements (Supplementary Material Section 2 platforms #7 and #8). Making 3-dimensional observations downwind of the platforms and using a sonic anemometer would help identify some of the unknowns presented here. Also, measuring more platforms over a longer time frame would improve the understanding of ambient leakage.

Any further measurements would be significantly easier with the cooperation of the oil and gas industry which could benefit from the findings. If the emissions are as low as the industry currently estimates, further measurements confirming low leakage rates would improve consumer confidence in oil and gas extraction activities. Alternately, if emissions are higher than currently reported, additional measurements would give the industry an opportunity to identify common issues such as incomplete combustion at the flare (Fig. 2), reduce leakage, and improve the efficiency of platforms thus potentially increasing profits from the extracted gas.

The continuous leakage of $CH_4$ from offshore production platforms observed here is consistent with observations of similar onshore operations (Omara et al., 2016; Riddick et al., 2019). Ambient leakage is not unexpected as these offshore production platforms are located in the inhospitable conditions of the North Sea, where wind speeds regularly exceed hurricane force and waves can reach the working deck. However, it is surprising that ambient leakage has not been explicitly factored into the UK national emissions inventory, which relies solely on operators self-reported emissions calculated using emission factors combined with specific processes like flaring. Without direct measurement, operators can remain unaware of small emissions that occur during normal operation.

The small amount of $CH_4$ lost as ambient leakage measured here may not be economically important, but when extrapolated to a global scale the loss of 0.19% of gas production (the production-weighted average loss) from offshore oil and gas production corresponds to a global emission of 0.8 Tg $CH_4$ $yr^{-1}$ (IEA, 2018). Currently, the Oil and Gas Climate Initiative (OGCI) estimates the global $CH_4$ emission from the oil and gas sector to be 1.6 Tg $CH_4$ $yr^{-1}$, based on the OGCI's own estimate that 0.32% of $CH_4$ extracted is lost (OGCI, 2018). This estimate represents data from 13 of the largest oil and gas producers and accounts for up-stream $CH_4$ emissions from flaring, venting and offshore oil loading for all operated gas and oil assets. If a global $CH_4$ emission from ambient leakage of 0.19% estimated by this study (0.8 Tg $CH_4$ $yr^{-1}$) is added to the current global estimate from flaring, venting and offshore oil loading (1.6 Tg $CH_4$ $yr^{-1}$) the total $CH_4$ emission from offshore oil and gas production would increase significantly. It should be noted that the value of 0.19% is based on a very small sample size using a method that comes with significant uncertainty. Moreover, the median value of this study (6.8 g $s^{-1}$) is much smaller than the regional median emission estimate of 99 g $s^{-1}$ for the Malay Peninsula and 15 g $s^{-1}$ for Borneo (Nara et al., 2015), which suggests that the ambient leakage rate may be lower in the North Sea than other regions of the world. This study does highlight the shortcomings of using emission factors which rely on *a-priori* knowledge of the source, in contrast with direct measurements that account for all emissions and better estimate total emissions. In conclusion, we suggest that additional measurements of offshore oil and gas production platform operations (e.g. Saudi Arabia, Brazil, Mexico, Norway and the

United States) be conducted to better inform leakage estimates and that these measurements be used to improve the UK and global $CH_4$ emission inventories.

## Author contributions

J. Staunton-Sykes, S. N. Riddick and N. R. P. Harris designed the experiment, S. N. Riddick, J. Staunton-Sykes, G. Allen, J. Pitt and G.L. Forster prepared equipment and calibrated the instruments, S. N. Riddick, J. Staunton-Sykes and G.L. Forster carried out the measurements and J. Staunton-Sykes, A. J. Manning, D. Lowry, and E. G. Nisbet provided analysis. D. L. Mauzerall and M. Celia were the project leaders and provided scientific oversight and guidance throughout the planning, implementation, collection, and data analysis processes. S. N. Riddick and D. L. Mauzerall wrote the paper with help from M. Celia, M. Kang and N. R. P. Harris and with contributions from all co-authors.

## Acknowledgements

The US National Oceanic and Atmospheric Administration Grant # AWD1004141 and the UK Natural Environment Research Council (NERC) through the Greenhouse gAs UK and Global Emissions (GAUGE) project on grant number NE/K002570/1 supported this research. We thank Glen Thistlethwaite (Ricardo) for his help understanding the BEIS Environmental and Emissions Monitoring System and the National Atmospheric Emissions Inventory. We also thank the reviewers for their valuable suggestions, and especially Reviewer 1 for the data provided.

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
