# Peer review of "Methane emissions from oil and gas platforms in the North Sea"

_Atmospheric Chemistry and Physics, 2019_

## Referee Comment (RC1) · Anonymous Referee #1 · 21 Mar 2019

General Comments

This study presents estimates of methane emission rates from oil and gas production platforms in the North Sea during normal operation without flaring. The authors collected and analyzed three types of data; 1) in-situ observations of CH4 mole fractions at Weymouth, UK; 2) samples of air collected in Tedlar bags and analyzed for CH4 mole fractions and carbon isotope ratios using continuous flow gas chromatography/isotope ratio mass spectrometer; and 3) shipboard in-situ observations of CH4 made circular transects around individual platforms. The in-situ observations of CH4 mixing ratios at Weymouth were only used for qualitatively. They analyzed the isotope data with a keeling plot to determine that enhancements had a major thermogenic component. They analyzed the shipboard in-situ observations with a Gaussian plume model. Anal-

ysis of the shipboard observations suggested methane emissions significantly higher than public inventories would suggest, but insufficient to explain the enhancements measured at Weymouth.

Overall, I think that this is paper has promise but needs major revision and possibly more work. I have reservations about: 1) the applicability of the Gaussian plume model to the conditions, though these reservations could well be ameliorated with additional information and author responses. 2) The relevance of the observations made at the coast 3) the quality of the results of the Keeling plot analysis 4) the uncertainty analysis of the Gaussian plume model

I may be missing it, but I can't find public access to much of the data in this paper. Best practices for reproducibility would have all data publicly accessible with access instructions given in the paper. I would appreciate the opportunity to inspect and analyze the data before making the recommendation to publish. There are a few places in the paper with insufficiently detailed information to fully understand the study - for example, what are the coordinates of the platforms and on what day was what platform observed? How much did winds vary over the course of the ship transects?

Specific Comments

Critique #1

The use of a Gaussian plume model requires careful consideration of the assumptions that go into such a model. Namely, the model assumes a homogenous, steady state flow with a steady point source.

I think that, for the most part, the conditions in this study satisfy those assumptions, but I do have the following reservations.

The Gaussian plume model employed by the authors assumes an infinitely high boundary layer and homogenous mixing throughout the boundary layer– which is to say that they include a reflection term at the surface, no reflection term at the top of the bound-

ary layer, and a uniform vertical mixing. This is a marine environment in a cool climate during the summer, and so a marine layer is likely. This would come with a very low boundary layer height and temperature inversion near the ocean surface. The emission heights are 50-70m and the inlet height is 2.5m. The assumptions of homogenous vertical mixing and no reflection off the top of the boundary layer are at risk.

I had code on hand to extract and plot meteorological fields from the GFS global forecast model archives (raw data obtained from https://ready.arl.noaa.gov/archives.php). I extracted boundary layer heights and winds at 1300UTC on the days of the campaign and plotted them below. Boundary layer height capped at 1500m in the plot for visibility. These data carry the caveat that GFS archive forecast data has error – particularly in the boundary layer height. I am happy to share these data/plots with the authors for their own use if they wish.

The paper includes a plot of the studied platforms, but no quantitative description of the locations (e.g., latitude and longitude, and which day each was measured). However, it does appear that the boundary layer height in the vicinity of the platforms may have been quite low, depending on when each was observed.

Critique #2

While the investigation that forms the bulk of the study is logically solid, the abstract begins with a line of reasoning that is quite circumstantial. It describes what motivated the study. While it is interesting to read about the authors motivation, excluding this information would make the definitive methods of the investigation clearer.

The passage is: "Recent studies suggest oil and natural gas production facilities in North America may be underestimating methane (CH4) emissions during extraction. This, coupled with unusually high CH4 mole fractions observed at coastal sites during onshore winds in the UK, suggests CH4 emissions from oil and gas extractions in the North Sea could be higher than previously reported."

I don't think the conclusion necessarily follows. Emissions can vary greatly between facilities and across production fields. The geology and technology used in the North American fields where the aforementioned studies were conducted is much different than those of the North Sea. Unusually high mole fractions observed at the coast when winds came from the sea do not necessarily point to emissions from the oil and gas industry. Airmass trajectories can be quite complicated and there are many sources on a continental scale.

If the authors want to include the in-situ observations at Weymouth in the paper, then they should include a trajectory analysis. The paper does work without this passage, though.

Critique #3

What is the uncertainty of the parameters of the linear model from your Keeling plot (Figure 2a)? What is the uncertainty in the source isotopic signature? From the appearance of the plot, there is a very poor correlation between observations and very high uncertainty in the source isotopic signature. It might be instructive to color the points by time. The data are likely insufficient to describe the source. It could just be that all the data were taken near each other in time, and so there is not enough variation in the CH4 concentration to extract a signal.

Critique #4

The description of the uncertainty analysis for the Gaussian plume model is lacking in detail, and there are some red flags. For one, the total uncertainty is given as +/-54% while the uncertainty due to stability class uncertainty alone is estimated at 54%, and the greatest source of uncertainty is said to be the emission height.

The uncertainty analysis does not explicitly define what is meant by "uncertainty". It is said "The overall uncertainty, calculated as the root of the sum of individual uncertainties squared...". This implies that the uncertainties are standard deviations of normally

distributed random errors. But the uncertainties are almost certainly correlated.

Technical Corrections

Page 1 The abstract should include a concise description of the methods

Page 2, Figure 1 Caption: Mention Weymouth observatory.

Page 3, Line 16: "To investigate the loss of CH4 from offshore oil and gas installations we use two approaches; 1) determine whether the source of CH4 enhancements at WAO could be from oil and gas production platforms; and 2) estimate an average CH4 loss from offshore installations by making direct measurements of CH4 emissions from off-shore production platforms in the North Sea". The listed items don't seem to be "approaches" to investigating the loss of CH4. Is this a typographical error? It would be nice to see this information replaced with a clear and concise description of the methods used in the paper.

Page 3, Line 24: "between 10:00 and 13:00" please include time zone.

Page 4, Line 6: "Measurements from boats of CH4 emissions from individual production platforms..." Careful, the CH4 mole ratio was measured and the emission rate was estimated using a simple model. I think it's a reach to say that the emissions were measured. The previous sentence uses my preferred language.

Page 5, Line 7: "The gas is considered to be well-mixed within the volume of the cone" This is an inaccurate description of the Gaussian Plume model. A Gaussian Plume describes the distribution of the mass of the gas at a given time as a multivariate Gaussian in space. To say that the gas is well mixed within a volume would suggest a uniform distribution in a finite region.

Page 6, Figure 2a): The correlation here looks very weak.

Page 8, Figure 3: Can this figure this be a plate showing all the observations rather than just the observations from 1 platform? I'm assuming the arrow shows the average

wind speed and direction? How much variability was there?

Page 9, Line 4: "The main uncertainty using the Gaussian plume approach in this study is in estimating the height of emission..." I would argue that there are many large sources of uncertainty in the Gaussian plume model, some of which are almost certainly greater than error in the emission height. For example, the assumption of homogenous diffusion.

2017−06−06 13:00:00

**Fig. 1.**

2017−07−05 13:00:00

**Fig. 2.**

[Figure]

2017−08−16 13:00:00

**Fig. 3.**

2017−08−24 13:00:00

**Fig. 4.**

2017−08−25 13:00:00

Wind Speed (m/s)
30
20
10
0

PBLH (m)
1500
1000
500
0

Longitude

Latitude

**Fig. 5.**

---

## Referee Comment (RC2) · Daniel Varon (Referee) · 21 Mar 2019

GENERAL COMMENTS:

The study describes the quantification of methane emissions from eight offshore oil and gas production platforms in the North Sea during summer 2017. The motivation for the study is that (1) anomalously high coastal methane enhancements have been observed in the UK under meteorological conditions that imply possible transport of emissions from the North Sea, (2) previous research indicates that methane emissions from oil and gas production facilities are often underestimated in bottom-up inventories, and (3) methane emissions from offshore oil and gas platforms are poorly constrained. The authors use isotopic analysis to confirm that the high UK coastal enhancements

are thermogenic in origin. In conjunction with a Gaussian plume model, they use in situ boat-based measurements of near-surface (2m) methane concentrations and meteorology to estimate emission rates from the eight offshore facilities. Key conclusions are that (1) the measurements of methane emissions from the platforms do not explain the anomalously high UK coastal methane enhancements, (2) the platform emissions are larger (as a fraction of gas production) than reported by the UK Department for Business, Energy, and Industrial Strategy (BEIS), (3) the emissions are associated not with operating processes like venting and flaring, but with ambient leakage, which is not included in the UK National Atmospheric Emission Inventory (NAEI), and (4) this leakage could make a significant contribution to the total emission of methane from global oil and gas operations.

My view is that the study is scientifically sound, clearly written, and wholly within the scope of ACP. It contributes to a well-documented and growing body of work on the quantification of methane emissions from individual point sources based on in situ and remote sensing observations of local methane concentrations. The authors present measurements of a small sample (N = 8) of offshore platforms, but they draw valuable conclusions nonetheless and rightly call for further study of these facilities. I would recommend acceptance of the manuscript for publication in ACP after minor revision based on the specific comments and technical corrections below.

SPECIFIC COMMENTS:

Page 2, Lines 13-14: To my knowledge, the studies cited (Zavala-Araiza et al., 2015 and Schwietzke et al., 2017) do not directly discuss methane emissions from offshore platforms. Perhaps this sentence could be broken into two sentences or parts, the first citing these studies as evidence that public inventories often underestimate methane emissions, and the second suggesting that the same may be true for offshore oil and gas platforms.

P2, L17-19: Nara et al. (2014) quantified methane emissions from offshore platforms

in Southeast Asia using a mass balance approach, but the authors describe that study as qualitative rather than quantitative. It would be helpful to clarify this comparison in the manuscript.

P3, L19-20: I would recommend removing this novelty claim, because the study is clearly original. Targeted measurement of methane emissions from individual oil and gas platforms is an impressive contribution. This sentence could be replaced with a one- or two-sentence comparison to the previous work of Nara et al. (2014).

P4, L24: The maximum horizontal distance from the platforms is reported to be 1500 m, but some platforms in Table 1 have distances of 2000 m.

P8, Figure 3: Peak enhancements (2160-2230 ppb) do not match the value reported in Table 1 (2290 ppb). Can the authors clarify in the table caption (or elsewhere in the manuscript) whether the downwind methane concentrations reported in Table 1 represent peak concentrations, or something else?

P8, L11-14: The total emission from the 8 platforms should not be compared to the total production from only 6 platforms unless there is good reason to believe that the missing production rates are small. Indeed, if one of platforms #1 or #2 produced as much gas as platform #4, the calculation would be quite different. One solution to this problem would be to compare emissions and production rates only for platforms #3-#8. Another option would be to impute the production rates for platforms #1 and #2 from the average (or median) of the other platforms' rates.

P9, L12-15: Why might the Pasquill-Gifford stability classes used to infer emissions from the platforms be too stable? What would cause the difference between stability at the receptor and stability at the source? Is it the difference in wind speed between the surface and 40-90 m altitude? If so, would this not suggest that the stability class as assessed at the surface might be too unstable (due to the winds being faster at altitude)? One additional sentence would probably clear this up.

[Figure]

P10, L8: Why are the estimated platform emissions larger than BEIS reported emissions of 0.13% by a factor of 2, but similar in magnitude to NAEI emissions? From page 6, line 1, it seems like the BEIS and NAEI figures should be similar, since the BEIS data "form the basis for emissions reported under category 1B2 within the National Atmospheric Emissions Inventory (NAEI; BEIS, 2018)." This can also probably be clarified in a sentence.

P10, L25-31: I am a bit hesitant to draw broad conclusions about global methane emissions from the oil and gas sector based on results from a small number of offshore platforms. It is interesting that the Oil and Gas Climate Initiative does not include ambient emissions in its global estimates when these emissions seem to be significant (as the authors illustrate), but I would expect their magnitude to vary greatly across geographies and industries. Indeed, the authors make note of this variability on page 2, line 15, and mention also the particularly harsh environment of the North Sea on page 10, line 20. I would recommend that the authors more clearly qualify their extrapolation of ambient emissions from North Sea offshore platforms to ambient emissions from global oil and gas activities.

TECHNICAL CORRECTIONS:

Page 1, Line 4: The words "onshore" and "offshore" are spelled differently throughout the text, both with and without dashes.

P2, L10: The acronym "OGA" is not defined.

P3, L11: The acronym "EEMS" is not defined.

P7, L10-11: Redundant use of the word "example."

P9, L8: It seems like there might be a missing word here.

---

## Author Comment (AC1) · 18 Jun 2019

Ms. Ref. No.: acp-2019-90

Title: Measuring methane emissions from oil and gas platforms in the North Sea

Department of Civil and Environmental Engineering
Princeton University
E320 Engineering Quad
Princeton
NJ

Email: sriddick@princeton.edu

18[th] June 2019

Dear Editor,

We thank Reviewer #1 for the comments and additional data supplied. As suggested, we have amended the manuscript to address the reviewers' comments and suggestions.

Please find our detailed responses below.

Yours sincerely,

Stuart Riddick (corresponding author)

Stuart N. Riddick, Denise L. Mauzerall, Michael Celia, Neil R. P. Harris, Grant Allen, Joseph Pitt, John Staunton-Sykes, Grant L. Forster, Mary Kang, David Lowry, Euan G. Nisbet, and Alistair J. Manning

**Anonymous Referee #1**

*General comment*
*Reviewer Comment:*
*I have reservations about: 1) the applicability of the Gaussian plume model to the conditions, though these reservations could well be ameliorated with additional information and author responses. 2) The relevance of the observations made at the coast 3) the quality of the results of the Keeling plot analysis 4) the uncertainty analysis of the Gaussian plume model*
*I may be missing it, but I can't find public access to much of the data in this paper. Best practices for reproducibility would have all data publicly accessible with access instructions given in the paper. I would appreciate the opportunity to inspect and analyze the data before making the recommendation to publish. There are a few places in the paper with insufficiently detailed information to fully understand the study - for example, what are the coordinates of the platforms and on what day was what platform observed? How much did winds vary over the course of the ship transects?*

Author's response:
In the responses below we address the reviewer's reservations regarding the suitability of the Gaussian plume model and the uncertainties. We also acknowledge the reviewer's concerns regarding the uncertainty analysis and have enhanced the content of this section to indicate the shortcomings of the measurement methodologies. To address the issue of data transparency we have introduced all data required to reproduce the emission estimates in Supplementary Materials Sections 1 and 2 and included latitudes and longitudes of the measured platforms.

With regard to the relevance of the observations made at the coast and the quality of the results of the Keeling plot analysis we have decided to remove these sections completely and make the focus of this manuscript solely on offshore measurements.

*Critique #1*
*The use of a Gaussian plume model requires careful consideration of the assumptions that go into such a model. Namely, the model assumes a homogenous, steady state flow with a steady point source.*
*I think that, for the most part, the conditions in this study satisfy those assumptions, but I do have the following reservations.*
*The Gaussian plume model employed by the authors assumes an infinitely high boundary layer and homogenous mixing throughout the boundary layer which is to say that they include a reflection term at the surface, no reflection term at the top of the boundary layer, and a uniform vertical mixing. This is a marine environment in a cool climate during the summer, and so a marine layer is likely. This would come with a very low boundary layer height and temperature inversion near the ocean surface. The emission heights are 50-70m and the inlet height is 2.5m. The assumptions of homogenous vertical mixing and no reflection off the top of the boundary layer are at risk.*
*I had code on hand to extract and plot meteorological fields from the GFS global forecast model archives (raw data obtained from https://ready.arl.noaa.gov/archives.php). I extracted boundary layer heights and winds at 1300UTC on the days of the campaign and plotted them below. Boundary layer height capped at 1500m in the plot for visibility. These data carry the caveat that GFS archive forecast data has error – particularly in the boundary layer height. I am happy to share these data/plots with the authors for their own use if they wish.*

*The paper includes a plot of the studied platforms, but no quantitative description of the locations (e.g., latitude and longitude, and which day each was measured). However, it does appear that the boundary layer height in the vicinity of the platforms may have been quite low, depending on when each was observed.*

**Author's response**
We thank the reviewer for highlighting issues with the Gaussian plume model and in particular for assisting our analysis by sharing data. We greatly appreciate these contributions and are confident that our additions have improved the manuscript. To address the reviewers concerns we have included the assumptions we have made when using the Gaussian plume model.

To specifically address the concerns over the plume's reflectance at the top of the boundary layer we have modified the Gaussian Plume model that we use to include terms that account for reflectance at the top of the boundary layer. The height of the boundary layer is taken from the Global Forecast System's global forecast model archives provided by the reviewer. These data are presented in the full data table in Supplementary Material Section 1, where all of the data used to calculate emissions from each platform can be found.

To address the issue of homogeneous mixing the shortcomings of the Gaussian plume model have been added to new sections: 2.2.5 Uncertainties; and 3.5 Uncertainties/shortcomings of Gaussian Plume modelling.

Change to manuscript:

[revised manuscript text omitted]

*Critique #2*
*While the investigation that forms the bulk of the study is logically solid, the abstract begins with a line of reasoning that is quite circumstantial. It describes what motivated the study. While it is interesting to read about the authors motivation, excluding this information would make the definitive methods of the investigation clearer.*
*The passage is: "Recent studies suggest oil and natural gas production facilities in North America may be underestimating methane (CH4) emissions during extraction. This, coupled with unusually high CH4 mole fractions observed at coastal sites during onshore winds in the UK, suggests CH4 emissions from oil and gas extractions in the North Sea could be higher than previously reported."*
*I don't think the conclusion necessarily follows. Emissions can vary greatly between facilities and across production fields. The geology and technology used in the North American fields where the aforementioned studies were conducted is much different than those of the North Sea. Unusually high mole fractions observed at the coast when winds came from the sea do not necessarily point to emissions from the oil and gas industry. Airmass trajectories can be quite complicated and there are many sources on a continental scale.*
*If the authors want to include the in-situ observations at Weymouth in the paper, then they should include a trajectory analysis. The paper does work without this passage, though.*

Author's response:
As suggested by the reviewer, using coastal observations to infer large emissions from offshore oil and gas operations is very difficult. The initial intention of the isotopic analysis of air collected during a Northerly wind event was to test if the air could have come from production platforms, i.e. has the isotopic signature as gas collected offshore. Unfortunately, as the analysis progressed it became apparent that these measurements could not be used to definitively identify the source.

To address these concerns in the manuscript we have removed the onshore/iostopic observations, the first sentence in the abstract has been removed and the second sentence edited.

Change to manuscript:
At P1 L17:
"To investigate whether offshore oil and gas platforms leak methane ($CH_4$) during normal operation, we measured $CH_4$ mole fractions around eight oil and gas production platforms in the North Sea which were neither flaring gas nor off-loading oil."

*Critique #3*
*What is the uncertainty of the parameters of the linear model from your Keeling plot (Figure 2a)? What is the uncertainty in the source isotopic signature? From the appearance of the plot, there is a very poor correlation between observations and very high uncertainty in the source isotopic signature. It might be instructive to color the points by time. The data are likely insufficient to describe the source. It could just be that all the data were taken near each other in time, and so there is not enough variation in the CH4 concentration to extract a signal.*

Author's response:
The reviewer's concerns over the isotopic analysis are justified. We feel that while this analysis could indicate offshore leakage, it does not necessarily compliment the central message of the manuscript, methane leaks from all offshore platforms. Instead of attempting to further justify the inclusion of the isotopic analysis to this body of work, we have decided to remove the section completely.

*Critique #4*
*The description of the uncertainty analysis for the Gaussian plume model is lacking in detail, and there are some red flags. For one, the total uncertainty is given as +/-54% while the uncertainty due to stability class uncertainty alone is estimated at 54%, and the greatest source of uncertainty is said to be the emission height.*
*The uncertainty analysis does not explicitly define what is meant by "uncertainty". It is said "The overall uncertainty, calculated as the root of the sum of individual uncertain- ties squared...". This implies that the uncertainties are standard deviations of normally distributed random errors. But the uncertainties are almost certainly correlated.*

Author's response:
New sections have been added to present and discuss the uncertainties in Gaussian Plume modelling.

Changes to the manuscript:
P6 L10

**"2.5 Uncertainties**

Of the Gaussian plume model assumptions presented in Section 2.3, two may not be valid - uniform vertical mixing and a constant wind speed. The uncertainty in uniform vertical mixing is discussed in Section 3.4. To investigate how uncertainties in the measurements and modelling affect the calculated emission, we ran Gaussian plume model scenarios using data that reflect the input values' uncertainty bounds. The scenarios run using the Gaussian plume approach were: varying wind speed (based on measurement); UGGA precision ($\pm$ 2 ppb); thermometer precision ($\pm$ 0.1 °C); the PGSC (+ 1 PGSC); and distance from detector to emission source ($\pm$ 50 m). The uncertainties of the UGGA and thermometer were taken from literature. The uncertainty in the PGSC used reflects the possibility that the temperature of the natural gas leaving the subsurface could be hotter than air and therefore less stable. The uncertainty in distance from the emission source to the detector results from not knowing where gas is leaking; here we assume the leak could be from anywhere on a production platform that is 100 m long.**"**

*Technical correction #1*
*Page 1 The abstract should include a concise description of the methods*

Author's reply:
A concise description of the methods has been included in the abstract

Change to manuscript:

P1 L19:

"We use the measurements from summer 2017, along with meteorological data, in a Gaussian plume model to estimate $CH_4$ emissions from each platform."

*Technical correction #2*
*Page 2, Figure 1 Caption: Mention Weymouth observatory.*

Author's reply:
The caption in Figure 1 has been edited

Change to manuscript:
P2 L14:
"**The map also shows the location of the University of East Anglia's Weybourne Atmospheric Observatory (WAO; 52.95 °N, 1.14 °E) in Weybourne, Norfolk.**"

*Technical correction #3*
*Page 3, Line 16: "To investigate the loss of CH4 from offshore oil and gas installations we use two approaches; 1) determine whether the source of CH4 enhancements at WAO could be from oil and gas production platforms; and 2) estimate an average CH4 loss from offshore installations by making direct measurements of CH4 emissions from off-shore production platforms in the North Sea". The listed items don't seem to be "approaches" to investigating the loss of CH4. Is this a typographical error? It would be nice to see this information replaced with a clear and concise description of the methods used in the paper.*

Author's reply:
As suggested a brief description of the methods has been included.

Changes to manuscript:
P3 L23:
"To investigate the $CH_4$ loss from offshore oil and gas installations in UK waters, we measure $CH_4$ mixing ratios downwind of offshore platforms and use these data in a Gaussian plume model to estimate $CH_4$ emission rates. The $CH_4$ loss is then presented as a percentage of the $CH_4$ produced by each platform."

*Technical correction #4*
*Page 3, Line 24: "between 10:00 and 13:00" please include time zone.*

Author's response:
This section has been removed

*Technical correction #5*
*Page 4, Line 6: "Measurements from boats of CH4 emissions from individual production platforms..." Careful, the CH4 mole ratio was measured and the emission rate was estimated using a simple model. I think it's a reach to say that the emissions were measured. The previous sentence uses my preferred language.*

Author's reply:
Corrected as suggested.

Change to manuscript:
P3 L30

"Measurements were made during normal operation (i.e. pilot light on the flare stack was lit, but no flaring or offshore oil loading was observed)"

*Technical correction #6*
*Page 5, Line 7: "The gas is considered to be well-mixed within the volume of the cone" This is an inaccurate description of the Gaussian Plume model. A Gaussian Plume describes the distribution of the mass of the gas at a given time as a multivariate Gaussian in space. To say that the gas is well mixed within a volume would suggest a uniform distribution in a finite region.*

Author's reply:
As suggested this line has been removed.

*Technical correction #6*
*Page 6, Figure 2a): The correlation here looks very weak.*

Author's reply:
As the reviewer notes, the results from this measurement are ambiguous. To address this we have removed the section.

*Technical correction #7*
*Page 8, Figure 3: Can this figure this be a plate showing all the observations rather than just the observations from 1 platform? I'm assuming the arrow shows the average wind speed and direction? How much variability was there?*

Author's reply:
All observations have been added as Supplementary Material Section 2.

During the measurements the highest uncertainty in wind speed during measurement was measured at ± 0.6 m s$^{-1}$ during the measurement of platform ID # 3 on the 6$^{th}$ July 2017.

P9 L14

"The largest variability in wind speed was recorded during measurement of platform # 3 on the 6$^{th}$ July 2017 at 4.4 ± 0.6 m s$^{-1}$ (Supplementary Material Section 1), using this variability in wind speed in the Gaussian plume model results in an uncertainty in average emission of ± 12 %."

*Technical correction #8*
*Page 9, Line 4: "The main uncertainty using the Gaussian plume approach in this study is in estimating the height of emission..." I would argue that there are many large sources of uncertainty in the Gaussian plume model, some of which are almost certainly greater than error in the emission height. For example, the assumption of homogenous diffusion.*

Author's reply:

As suggested by the reviewer the section describing the uncertainties has been changed to include a systematic estimate of major uncertainties.

Changes to the manuscript:

P9 L10

**"3.4 Uncertainties/shortcomings of Gaussian Plume modelling**

A range of scenarios were run using the Gaussian plume model to estimate uncertainty in average $CH_4$ emissions resulting from UGGA instrument precision, thermometer precision, varying wind speed, assessment of the PGSC and uncertainty in distance between the emission source and the detector. Uncertainty in the UGGA and the thermometer have little effect on the average emission estimate (Supplementary Material Section 3). The largest variability in wind speed was recorded during measurement of platform # 3 on the 6th July 2017 at $4.4 \pm 0.6$ m s$^{-1}$ (Supplementary Material Section 1), using this variability in wind speed in the Gaussian plume model results in an uncertainty in average emission of $\pm 12$ %. Uncertainty in estimating the distance between the emission source and the detector results in an uncertainty in average emissions of $\pm 8$ %. The Gaussian plume model has the greatest response to the uncertainty in estimating the PGSC, resulting in an uncertainty of $\pm 41$ %. We estimate the overall uncertainty in the average $CH_4$ emission, calculated as the root of the sum of the individual uncertainties squared, to be $\pm 45$ %.

As mentioned in Section 2.5, the uniform vertical mixing assumption made in the Gaussian Plume model may not hold here as the data we collected provides no information on vertical mixing. However, the Gaussian plume model only assumes a constant vertical mixing rate between the source and the detector. In most cases this distance is relatively short and unlikely to significantly affect the calculation of emissions. In future experiments, the vertical mixing rate could be calculated by measuring the vertical gradient of wind speeds to make an accurate thermodynamic profile."

P10 L17
"The emission estimates presented here are from a pilot study and further work is needed to establish total $CH_4$ leakage rates from offshore oil and gas platforms. We have established, however, that $CH_4$ enhancements can be detected downwind of all production platforms during normal operations when neither venting, flaring or oil loading activities are taking place. Our measurements used in a Gaussian plume model indicate leakage from offshore installations are likely larger than previously estimated. However, these emission estimates come with large uncertainties as they are based on relatively few measured platforms, assume values for the height of emission, lateral and vertical mixing ratio distributions, and may not meet all the Gaussian plume model assumptions.
When the $CH_4$ emissions are calculated for two different emission heights, the importance of identifying the source location and height above the sea becomes apparent. The median $CH_4$ emissions from the five platforms is 6.8 g s$^{-1}$ when the emissions are all assumed to come from the working deck, while the median emissions is 2,658 g s$^{-1}$ (47% of production) when all $CH_4$ is assumed to be emitted from the flare i.e. the highest point of the platform. This analysis indicates that the median emission presented here, based on the assumption that the

emissions occur from the working deck, is a conservative estimate. However, without further measurements the height of the emission source cannot be definitively determined and this leaves the possibility that leakage is higher during normal operations than our results indicate. The other input variables that cannot be determined without further measurement are the lateral and vertical mixing ratio distributions but we feel that following the study of Blackall et al. (2007) the estimates used in this study are sufficient to establish leakage from oil and gas platforms and to provide a rough estimate of their emissions. As with the emission height, mixing can be resolved with further measurement, including the use of aircraft to resolve the vertical and horizontal mixing of the plume. It is clear that further studies are needed to provide additional data that will yield more definitive emission estimates. Using the near-source (< 1 km) observations of this paper (Fig. 2; Supplementary Material Section 2 platform #5 and #6) we can see that plumes from the leaks are compact (<200 m wide) and in some cases difficult to detect from sea-level measurements (Supplementary Material Section 2 platforms #7 and #8). Making 3-dimensional observations downwind of the platforms and using a sonic anemometer would help identify some of the unknowns presented here. Also, measuring more platforms over a longer time frame would improve the understanding of ambient leakage.

Any further measurements would be significantly easier with the cooperation of the oil and gas industry which could benefit from the findings. If the emissions are as low as the industry currently estimates, further measurements confirming low leakage rates would improve consumer confidence in oil and gas extraction activities. Alternately, if emissions are higher than currently reported, additional measurements would give the industry an opportunity to identify common issues such as incomplete combustion at the flare (Fig. 2), reduce leakage, and improve the efficiency of platforms thus potentially increasing profits from the extracted gas."

---

## Author Comment (AC2) · 18 Jun 2019

Ms. Ref. No.:  acp-2019-90

Title: Measuring methane emissions from oil and gas platforms in the North Sea

Department of Civil and Environmental Engineering
Princeton University
E320 Engineering Quad
Princeton
NJ

Email: sriddick@princeton.edu

18th June 2019

Dear Editor,

We thank Mr Varon for his comments. As suggested, we have amended the manuscript to address the reviewers' comments and suggestions.

Please find our detailed responses below.

Yours sincerely,

Stuart Riddick (corresponding author)

Stuart N. Riddick, Denise L. Mauzerall, Michael Celia, Neil R. P. Harris, Grant Allen, Joseph Pitt, John Staunton-Sykes, Grant L. Forster, Mary Kang, David Lowry, Euan G. Nisbet, and Alistair J. Manning

**Reviewer: Daniel Varon**

*Specific comment #1*
*Page 2, Lines 13-14: To my knowledge, the studies cited (Zavala-Araiza et al., 2015 and Schwietzke et al., 2017) do not directly discuss methane emissions from offshore platforms. Perhaps this sentence could be broken into two sentences or parts, the first citing these studies as evidence that public inventories often underestimate methane emissions, and the second suggesting that the same may be true for offshore oil and gas platforms.*

Author's response:
As suggested this sentence has been edited.

Change to manuscript:
P2 L19
"However, recent studies indicate public inventories in the United States underestimate $CH_4$ emissions

including from the oil and gas supply chain (Alvarez et al., 2018; Zavala-Araiza et al., 2015, Schwietzke et al.,

2017). This leads to the question: Could $CH_4$ emissions from offshore oil and gas platforms be higher than

previously estimated?"

*Specific comment #2*
*P2, L17-19: Nara et al. (2014) quantified methane emissions from offshore platforms in Southeast Asia using a mass balance approach, but the authors describe that study as qualitative rather than quantitative. It would be helpful to clarify this comparison in the manuscript.*

Author's reply:
As suggested a comparison between the findings of this study and Nara et al. (2014) have been made and included in the introduction and the discussion.

Changes to manuscript:
P3 L6
"However, a mass-balance approach identifies $CH_4$ emissions from offshore oil and gas operations off the coast

of South East Asia as having a large regional median (range) emission of 99 (4 – 427) g $CH_4$ $s^{-1}$ platform$^{-1}$ for the

Malay Peninsula and 15 (2 – 46) g $CH_4$ $s^{-1}$ platform$^{-1}$ for Borneo (Nara et al., 2015)."

P11 L29
"Moreover, the median value of this study (6.8 g $s^{-1}$) is much smaller than the regional median emission estimate

of 99 g $s^{-1}$ for the Malay Peninsula and 15 g $s^{-1}$ for Borneo (Nara et al., 2015), which suggests that the ambient

leakage rate may be lower in the North Sea than other regions of the world."

*Specific comment #3*
*P3, L19-20: I would recommend removing this novelty claim, because the study is clearly original. Targeted measurement of methane emissions from individual oil and gas platforms is an impressive contribution. This sentence could be replaced with a one- or two-sentence comparison to the previous work of Nara et al. (2014).*

Author's comment:
Thank you for the endorsement of our work. As suggested, we have removed the sentence about novelty.

*Specific comment #4*
*P4, L24: The maximum horizontal distance from the platforms is reported to be 1500 m, but some platforms in Table 1 have distances of 2000 m.*

Author's reply:
This was a typo and has been corrected.

Change to manuscript:
P4 L21
"and 2 km horizontal distance"

*Specific comment #5*
*P8, Figure 3: Peak enhancements (2160-2230 ppb) do not match the value reported in Table 1 (2290 ppb). Can the authors clarify in the table caption (or elsewhere in the manuscript) whether the downwind methane concentrations reported in Table 1 represent peak concentrations, or something else?*

Author's response:
This should have been platform ID #6 instead of # 5. The caption has been corrected.

Change to manuscript:
P8 Caption Figure 2
"Minute-averaged $CH_4$ mole fraction measurements made upwind and downwind of production platform, ID # 6, on the 24th of August."

*Specific comment #6*
*P8, L11-14: The total emission from the 8 platforms should not be compared to the total production from only 6 platforms unless there is good reason to believe that the missing production rates are small. Indeed, if one of platforms #1 or #2 produced as much gas as platform #4, the calculation would be quite different. One solution to this problem would be to compare emissions and production rates only for platforms #3-#8. Another option would be to impute the production rates for platforms #1 and #2 from the average (or median) of the other platforms' rates.*

Author's response:
As suggested we have changed the calculation to only include platforms 3 to 8. Text has been included to the caption of Table 1 and the manuscript to reflect this.

Change to manuscript:
P7 Table 1 caption
"**The calculation of the "Median", "Mean" and "Total" only use data from platforms #4 to # 8. Platforms #1 and #2 did not have production data available for the time of measurement. During the measurement of Platform #3 the height of the PBL was calculated as zero (GFS, 2019) making the Gaussian plume modelled emission estimate ambiguous.**"

P8 L11

"During the measurement of platform #3, the calculated boundary layer height was 0 m (GFS, 2019) making the emission estimate ambiguous and, even though presented in Table 1, has not been used further in the analysis. Using emission data from the five platforms with available production data and with a non-zero calculated PBL (platforms #4 through #8), the median $CH_4$ emission was 6.8 g $s^{-1}$ (mean 11.2 g $s^{-1}$)."

*Specific comment #7*
*P9, L12-15: Why might the Pasquill-Gifford stability classes used to infer emissions from the platforms be too stable? What would cause the difference between stability at the receptor and stability at the source? Is it the difference in wind speed between the surface and 40-90 m altitude? If so, would this not suggest that the stability class as assessed at the surface might be too unstable (due to the winds being faster at altitude)? One additional sentence would probably clear this up.*

Author's response:
The methane lost from the platform may be less stable as it has come from the subsurface and may be a warmer than the surrounding air and therefore less stable. As a test, we suggest this could be 1 PGSC less stable than calculated. To clarify this text has been added.

Change to manuscript:
P6 L11
"The uncertainty in the PGSC used reflects the possibility that the temperature of the natural gas leaving the subsurface could be hotter than air and therefore less stable."

*Specific comment #8*
*P10, L8: Why are the estimated platform emissions larger than BEIS reported emissions of 0.13% by a factor of 2, but similar in magnitude to NAEI emissions? From page 6, line 1, it seems like the BEIS and NAEI figures should be similar, since the BEIS data "form the basis for emissions reported under category 1B2 within the National Atmospheric Emissions Inventory (NAEI; BEIS, 2018)." This can also probably be clarified in a sentence.*

Author's reply:
The platform emissions are twice as large as the BEIS emission estimates but appear to be consistent with the NAEI because NAEI currently only accounts for venting and flaring not leakage. Here we present the leakage estimates only as venting and flaring were not taking place. It is only by coincidence that our leakage estimates are the same as the NAEI values.

Change to manuscript:
P10 L11
"neither of which was taking place during our measurements"

*Specific comment #9*
*P10, L25-31: I am a bit hesitant to draw broad conclusions about global methane emissions from the oil and gas sector based on results from a small number of offshore platforms. It is interesting that the Oil and Gas Climate Initiative does not include ambient emissions in its global estimates when these emissions seem to be significant (as the authors illustrate), but I would expect their magnitude to vary greatly across geographies and industries. Indeed, the authors make note of this variability on page 2, line 15, and mention also the particularly harsh environment of the North Sea on page 10, line 20. I would recommend that the authors more*

*clearly qualify their extrapolation of ambient emissions from North Sea offshore platforms to ambient emissions from global oil and gas activities.*

Author's reply:
The text has been amended to reflect the speculative nature of this statement. The idea of this paragraph was to merely represent the concept of emissions from leakage and the potential impact of these measurements.

Change to manuscript:
P11 L25
"If a global $CH_4$ emission from ambient leakage of 0.19% estimated by this study (0.8 Tg $CH_4$ $yr^{-1}$) is added to the current global estimate from flaring, venting and offshore oil loading (1.6 Tg $CH_4$ $yr^{-1}$) the total $CH_4$ emission from offshore oil and gas production would increase significantly. It should be noted that the value of 0.19% is based on a very small sample size using a method that comes with significant uncertainty."

*Technical correction #1*
*Page 1, Line 4: The words "onshore" and "offshore" are spelled differently throughout the text, both with and without dashes.*

Author's reply:
Have amended to be consistently "onshore" and "offshore".

*Technical correction #2*
*P2, L10: The acronym "OGA" is not defined.*

Author's reply
The acronym has been defined as the UK Oil and Gas Authority.

Change to MS:
P2 L16:
"(UK Oil and Gas Authority, 2018)."

*Technical correction #3*
*P3, L11: The acronym "EEMS" is not defined.*

Author's reply:
EEMS has been defined as the Environmental and Emissions Monitoring System.

Change to MS:
P3 L17:
"UK Government's Department of Energy and Climate Change Environmental and Emissions Monitoring System (DECC EEMS, 2008)"

*Technical correction #4*
*P7, L10-11: Redundant use of the word "example."*

Author's reply:
Deleted as suggested.

*Technical correction #5*
*P9, L8: It seems like there might be a missing word here.*

Author's reply:
Have amended the sentence.

Change to MS:
P8 L14
"As a sensitivity study, the median modelled emission is 2,658 g s$^{-1}$ (mean 1,892 g s$^{-1}$) when we assume all $CH_4$

is emitted from the highest point of the platform, i.e. the flare."